# PRINCIPLED HYBRIDS OF GENERATIVE AND DISCRIMINATIVE DOMAIN ADAPTATION

## ABSTRACT

We propose a probabilistic framework for domain adaptation that blends both generative and discriminative modeling in a principled way. Under this framework, generative and discriminative models correspond to specific choices of the prior over parameters. This provides us a very general way to interpolate between generative and discriminative extremes through different choices of priors. By maximizing both the marginal and the conditional log-likelihoods, models derived from this framework can use both labeled instances from the source domain as well as unlabeled instances from *both* source and target domains. Under this framework, we show that the popular reconstruction loss of autoencoder corresponds to an upper bound of the negative marginal log-likelihoods of unlabeled instances, where marginal distributions are given by proper kernel density estimations. This provides a way to interpret the empirical success of autoencoders in domain adaptation and semi-supervised learning. We instantiate our framework using neural networks, and build a concrete model, *DAuto*. Empirically, we demonstrate the effectiveness of DAuto on text, image and speech datasets, showing that it outperforms related competitors when domain adaptation is possible.

## 1 INTRODUCTION

Making accurate predictions relies heavily on the existence of labeled data for the desired tasks. However, generating labeled data for new learning tasks is often time-consuming. As a result, this poses an obstacle for applying machine learning methods to broader application domains. *Domain adaptation* focuses on the situation where we only have access to labeled data from source domain, which is assumed to be different from the target domain we want to apply our model to. The goal of domain adaptation algorithms under this setting is to generalize better in the target domain by exploiting labeled data in the source domain and unlabeled data in the target domain.

In this paper we propose a probabilistic framework for domain adaptation that combines both generative and discriminative modeling in a principled way. We start from a very simple yet general generative model, and show that a special choice on the prior distribution of model parameters leads to the usual discriminative modeling. Due to its generative nature, the framework provides us a principled way to use unlabeled instances from both the source and the target domains. Under this framework, if we use non-parametric kernel density estimators for the marginal distribution over instances, we can show that the popular reconstruction loss of autoencoders corresponds to an upper bound of the negative marginal log-likelihoods of unlabeled instances. This provides a novel probabilistic interpretation on why unsupervised training with general autoencoders may help with discriminative tasks, though interpretations exist for specific variants of autoencoders, e.g., denoising autoencoders (Vincent et al., 2008) and contractive autoencoders (Rifai et al., 2011). Our interpretation may also be used to explain the recent success of autoencoders in semi-supervised learning (Rasmus et al., 2015).

We instantiate our framework with flexible neural networks, which are powerful function approximators, leading to a concrete model, *DAuto*. DAuto is designed to achieve the following three objectives simultaneously in a unified model: 1). It learns representations that are informative for the main learning task in the source domain. 2). It learns domain-invariant features that are indistinguishable between the source and the target domains. 3). It learns robust representations under reconstruction loss for instances in both domains. To demonstrate the effectiveness of DAuto, we first conduct a

synthetic experiment using the MNIST dataset, showing its superior performance when adaptation is possible. We further compare DAuto with state-of-the-art models on the Amazon benchmark dataset. As another contribution, we extend DAuto so that it can also be applied in time-series modeling. We evaluate it on a speech recognition task, showing its effectiveness in improving recognition accuracies when trained from utterances with different accents. In the end, we also provide qualitative analysis in the case where domain adaptation is hard and all the methods we test fail.

## 2 RELATED WORK

Recently, due to the availability of rich data and powerful computational resources, non-linear representations and hypothesis classes for domain adaptation have been increasingly explored (Glorot et al., 2011; Baktashmotlagh et al., 2013; Chen et al., 2012; Ajakan et al., 2014; Ganin et al., 2016). This line of works focuses on building common and robust feature representations among multiple domains using either supervised neural networks (Glorot et al., 2011), or unsupervised pretraining using denoising auto-encoders (Vincent et al., 2008; 2010). Other works focus on learning feature transformations such that the feature distributions in the source and target domains are close to each other (Ben-David et al., 2007; 2010; Ajakan et al., 2014; Ganin et al., 2016). In practice it was observed that unsupervised pretraining using stacked denoising auto-encoders (mSDA) (Vincent et al., 2008; Chen et al., 2012) often improves the generalization accuracy (Ganin et al., 2016). One of the limitations of mSDA is that it needs to explicitly form the covariance matrix of input features and then solves a linear system, which can be computationally expensive to solve exactly in high dimensional settings, but approximate scheme exists.

Domain adversarial neural networks (DANN) is a discriminative model to learn domain-invariant features (Ganin et al., 2016). It can be formulated as a minimax problem where the feature transformation component tries to learn a representation to confuse a following domain classification component. DANN also enjoys a nice theoretical justification to learn a feature map to decrease the $\mathcal{A}$-distance measure (Ben-David et al., 2007) between source and target domains. Other distance measures between distributions can also be applied. Tzeng et al. (2014) and Long et al. (2015) propose similar models where the maximum mean discrepancy (MMD) (Gretton et al., 2012) between two domains are minimized. Tzeng et al. (2017) also proposed a variant of DANN, known as ADDA, where the encoders for source and target domains are not shared. Instead of joint training objective classifier and domain discriminator, ADDA runs in two stages. At the first stage, source encoder and classifier are trained in a supervised way on the source domain. At the second stage, both source encoder and source classifier are fixed, and both domain discriminator and target encoder are trained using only unsupervised samples from both domains. Very recently, Bousmalis et al. (2016) propose a model where orthogonal representations that are shared between domains and unique to each domain are learned simultaneously. They achieve this goal by incorporating both similarity and difference penalties for features into the objective function. Finally, domain adaptation can also be viewed as a semi-supervised learning problem by ignoring the domain shift, where source instances are treated as labeled data and target instances are unlabeled data (Dai et al., 2007; Rasmus et al., 2015).

DAuto improves over mSDA (Chen et al., 2012) when the dimension of feature vectors is high and as a result the covariance matrix cannot be explicitly formed. As we will see later, DAuto can also be naturally extended to modeling in time-series domain. On the other hand, compared with DANN, DAuto has a clear probabilistic generative model interpretation and provides us a principled way to use unlabeled data from both the source and target domains during training.

## 3 A PRINCIPLED HYBRID MODEL FOR DOMAIN ADAPTATION

Generative models provide a principled way to use both labeled data from source domain and unlabeled data from both domains. In this section we start with a general probabilistic framework for domain adaptation using a principled hybrid of both generative and discriminative models. We then provide a concrete instantiation of our framework, and show that it leads to popular reconstruction-based domain adaptation models. Our derivation can also be used to explain the prevalence and success of autoencoders in both domain adaptation and semi-supervised learning (Rasmus et al., 2015; Bousmalis et al., 2016). We end this section by proposing the DAuto model that follows

our probabilistic framework and combines the minimax adversarial loss as a regularizer for domain adaptation.

## 3.1 A Probabilistic Framework for Domain Adaptation

Let $\mathbf{x} \in \mathbb{R}^d$ be an input instance and $y$ be its target variable: $y \in \{0, 1\}$ in the classification setting or $y \in \mathbb{R}$ in the regression setting. A fully generative model can be specified as follows:

$$p(\mathbf{x}, y; \phi, \psi) = p(\mathbf{x}; \phi)p(y \mid \mathbf{x}; \psi)p(\phi, \psi) \tag{1}$$

where $\phi$ is the model parameter that governs the generation process of $\mathbf{x}$; $\psi$ is the model parameter for the conditional distribution $y \mid \mathbf{x}$, and $p(\phi, \psi)$ is the prior distribution over both model parameters. At first glance, one might find the above generative model unsuitable under the domain adaptation setting since it implicitly assumes that the marginal distribution $p(\mathbf{x}; \phi)$ is shared among all instances from both domains. The key point that makes the above framework still valid under domain adaptation lies in the possible richness of the generation process parametrized by $\phi$: although the marginal distribution may be different in the input space $\mathbb{R}^d$, we can still find proper transformation $f : \mathbb{R}^d \to \mathbb{R}^D$ such that the induced marginal distributions (by $f$) over both domains are similar in $\mathbb{R}^D$ (Fig. 1). In fact, this is a necessary and implicit assumption that underlies the recent success of regularization based discriminative models (Tzeng et al., 2014; Long et al., 2015; Ganin et al., 2016). To better understand this, we illustrate the generative process for $\mathbf{x}$ in Fig. 1.

Using the above joint model, if we assume that the prior distribution $p(\phi, \psi)$ factorizes as $p(\phi, \psi) = p(\phi)p(\psi)$, then we will have:

$$\max_{\phi, \psi} p(\mathbf{x}; \phi)p(y \mid \mathbf{x}; \psi)p(\phi)p(\psi) = \max_{\psi} p(y \mid \mathbf{x}; \psi)p(\psi) \cdot \max_{\phi} p(\mathbf{x}; \phi)p(\phi) \tag{2}$$

Note that in this case only the first term in R.H.S. in (2) is concerned with the prediction, which means unsupervised learning on $p(\mathbf{x}; \phi)p(\phi)$ does not help generalization on prediction. In other words, the independence assumption between $\phi$ and $\psi$ equivalently reduces our joint model over both $\mathbf{x}$ and $y$ into discriminative models that only contain parameters $\psi$ if we only care about prediction accuracy. We emphasize that in this reduction, the assumption $p(\phi, \psi) = p(\psi)p(\phi)$ is crucial, otherwise the optimal $\psi^*$ will still depend on $\phi$, i.e., $\psi^* = \psi^*(\phi)$. On the other extreme, if we have $\phi = \psi$, then this corresponds to having a prior $p(\phi, \psi) = p_0(\phi, \psi)\delta(\phi - \psi)$ that constrains $\phi$ and $\psi$ to be shared in both generative processes:

$$\max_{\phi, \psi} p(\mathbf{x}; \phi)p(y \mid \mathbf{x}; \psi)p_0(\phi, \psi)\delta(\phi - \psi) = \max_{\phi} p(\mathbf{x}; \phi)p(y \mid \mathbf{x}; \phi)p_0(\phi) \tag{3}$$

where $p_0$ is a base distribution and $\delta(\cdot)$ denotes the Kronecker delta function. It can be seen that when $\phi = \psi$, the formulation exactly reduces to the usual MAP inference scheme in a generative model over both $\mathbf{x}$ and $y$, given parameter $\phi$.

The above discussion shows that the formulation given in (1) is general enough to incorporate both discriminative and generative modelings as two extreme cases, and depending on the choice of the prior distribution over $\phi$ and $\psi$, we can easily recover both. In a nutshell, when $\phi$ and $\psi$ are independent, we recover the discriminative modeling; otherwise if $\phi$ and $\psi$ are exactly the same, we recover the generative modeling. However in practice the sweet spot often lies in a mix of both models (Ng & Jordan, 2002): discriminative training usually wins at predictive accuracy, while generative modeling provides a principled way to use unlabeled data. To achieve the best of both worlds, now let us consider the case where $\phi$ and $\psi$ have a common subspace, i.e., some model parameters are shared in both the generation process of $\mathbf{x}$ and $y \mid \mathbf{x}$. Clearly under this case the factorization assumption of the prior distribution does not hold anymore, and we cannot hope to recover a discriminative model by simply optimizing $\psi$. To make our discussion concrete, think if we have $p(\mathbf{x}; \phi) = g(f(\mathbf{x}; \zeta); \phi \backslash \zeta)$ and $p(y \mid \mathbf{x}; \psi) = h(f(\mathbf{x}; \zeta), y; \psi \backslash \zeta)$, where $\zeta$ are the shared parameters of both $\phi$ and $\psi$. Domain adaptation is possible under this setting whenever $f(\cdot; \zeta)$ forms a rich class of transformations so that unlabeled instances from both domains have similar induced marginal distribution. This is a necessary condition for domain adaptation to succeed under our framework. As a generative model, it also allows algorithms to use unlabeled instances from both domains to optimize the marginal likelihood function $p(\mathbf{x}; \phi)$, which also helps the predictive task $p(y \mid \mathbf{x}; \psi)$ due to the shared component $\zeta$.

## 3.2 An Instantiation using Kernel Density Estimation

Now we use our probabilistic framework and instantiate it with proper choices of both the marginal distributions as well as the conditional distributions. On one hand, we would like to make as few assumptions as possible about the generation process of $\mathbf{x}$; on the other hand, the model should be rich enough such that even though instances from source and target domains may have different distributions in $\mathbb{R}^d$, the model contains flexible transformation under which the induced distributions are similar enough in both domains. Taking both considerations into account, we propose to use nonparametric kernel density estimator (KDE) to model $p(\mathbf{x}; \phi)$. Specifically, let $K(\cdot)$ be the chosen kernel and $\{\mathbf{x}_i\}_{i=1}^n$ be a set of unlabeled instances. Our KDE for $p(\mathbf{x}; \phi)$ is given by:

$$p(\mathbf{x}; \phi) \propto \frac{1}{nw} \sum_{i=1}^n K\left(\frac{\mathbf{x} - g(f(\mathbf{x}_i; \zeta); \phi\backslash\zeta)}{w}\right) \tag{4}$$

where $w > 0$ is the bandwidth and $f : \mathbb{R}^d \to \mathbb{R}^D$ and $g : \mathbb{R}^D \to \mathbb{R}^d$ are two feature transformations. Our definition of KDE differs from the original one (Wassermann, 2006) by the additional parametric transformations $g \circ f$ applied to $\mathbf{x}$, and when $g \circ f = I$, the identity map, our definition reduces to the original one. Note that when applied to the source and target domains separately, the original KDE does not give similar density estimations if their empirical distributions are far from each other, which is exactly the case under domain adaptation.

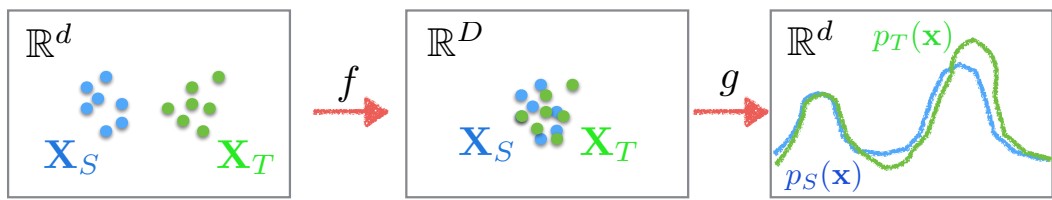

Figure 1: Density estimator with additional transformations $f$ and $g$. Input instances $\mathbf{X}_S$ and $\mathbf{X}_T$ are from source and target domains, respectively. $f$ transforms both $\mathbf{X}_S$ and $\mathbf{X}_T$ to $\mathbb{R}^D$ so that they are close in $\mathbb{R}^D$; $g$ transforms them back so that they have close density estimations in $\mathbb{R}^d$.

For the conditional distribution $y \mid \mathbf{x}$, depending on whether $y \in \mathbb{R}$ or $y \in \{0, 1\}$, typical choices include linear regression or logistic regression. While both these two models are linear and limited, we can first augment them with rich nonlinear transformation $f$ applied to the input instance.

Note that the transformation $f$, along with its parameters, are shared between both $p(\mathbf{x}; \phi)$ and $p(y \mid \mathbf{x}; \psi)$. Finally, our model is completed by specifying the prior distribution as follows:

$$p(\phi, \psi) = p_0(\phi, \psi)\delta(\phi(\zeta) - \psi(\zeta))$$

The $\delta(\cdot)$ constrains the common parameter $\zeta$ to be shared by both $p(\mathbf{x}; \phi)$ and $p(y \mid \mathbf{x}; \psi)$. The base distribution $p_0(\phi, \psi)$ can be chosen as a flat (possibly improper) prior, which corresponds to the usual MLE criterion; or other forms of distributions that effectively introduce regularizations on both $\phi$ and $\psi$.

## 3.3 Learning by Maximizing Joint Likelihood

One of the advantages a generative model lends us is a principled way of using unlabeled data from both domains. Let $\{(\mathbf{x}_i, y_i)\}_{i=1}^m$ be labeled data from source domain and $\{\mathbf{x}_j\}_{j=1}^n$ be unlabeled data from both domains. Instead of just maximizing the conditional likelihood using labeled data, we can jointly maximize both the conditional likelihood and the marginal likelihood:

$$\psi^*, \phi^* = \arg\min_{\psi, \phi} -\sum_{j=1}^n \log p(\mathbf{x}_j; \phi) - \sum_{i=1}^m \log p(y_i \mid \mathbf{x}_i; \psi) \tag{5}$$

It is clear that the negative conditional log-likelihood function is exactly the cross-entropy loss between the true label $y_i$ and its predicted counterpart. For the negative marginal likelihood, if we

choose a Gaussian kernel and plug in our KDE estimator given in (4), we have:

$$-\log p(\mathbf{x}_j; \phi) \propto -\log \left( \frac{1}{nw} \sum_{j'=1}^{n} \exp \left( -\frac{||\mathbf{x}_j - g(f(\mathbf{x}_{j'}; \zeta); \phi\backslash\zeta)||_2^2}{2w^2} \right) \right)$$

$$\leq -\log \frac{1}{nw} \exp \left( -\frac{||\mathbf{x}_j - g(f(\mathbf{x}_j; \zeta); \phi\backslash\zeta)||_2^2}{2w^2} \right)$$

$$= \lambda \cdot ||\mathbf{x}_j - g(f(\mathbf{x}_j; \zeta); \phi\backslash\zeta)||_2^2 + c \tag{6}$$

where $\lambda > 0$ only depends on the bandwidth and $c$ is a constant that does not depend on $\mathbf{x}_j$. The upper bound becomes tight as $w \to 0$. Note that in the above derivation we omit the prior distribution $p(\phi, \psi) = p_0(\phi, \psi)\delta(\phi(\zeta) - \psi(\zeta))$ by assuming that $p_0$ is a flat constant distribution. If other design choice is made, e.g., a Gaussian over $\phi(\psi)$, then there will be a corresponding $\ell_2$ regularization term for the model parameters $\phi(\psi)$. Putting all together, maximizing a combination of conditional and marginal likelihoods correspond to the following unconstrained minimization problem:

$$\text{minimize}_{\psi,\phi} \quad -\sum_{i=1}^{m} \log p(y_i \mid \mathbf{x}_i; \psi) + \lambda \sum_{j=1}^{n} ||\mathbf{x}_j - g(f(\mathbf{x}_j; \zeta); \phi\backslash\zeta)||_2^2 \tag{7}$$

**Remark**. The maximum of joint likelihood estimator is also the maximum-a-posteriori estimator, leading to a fully probabilistic interpretation of (7). The second term in the objective function has an interesting interpretation: it essentially measures the reconstruction error after a transformation by $g \circ f$. Based on (6), if we interpret $f$ as an encoder and $g$ as the corresponding decoder, then minimizing the reconstruction loss from an autoencoder exactly corresponds to maximizing a lower bound of the marginal probability distribution $p(\mathbf{x})$, where $p(\mathbf{x})$ is given by our kernel density estimation. Furthermore, the squared $\ell_2$ measure in the reconstruction error is not restricted: for example, if instead of a Gaussian kernel we chose the Laplacian kernel, then we would have an $\ell_1$ measure of the reconstruction loss in (7). This interpretation may also be used to explain the practical success of autoencoders in semi-supervised learning (Rasmus et al., 2015).

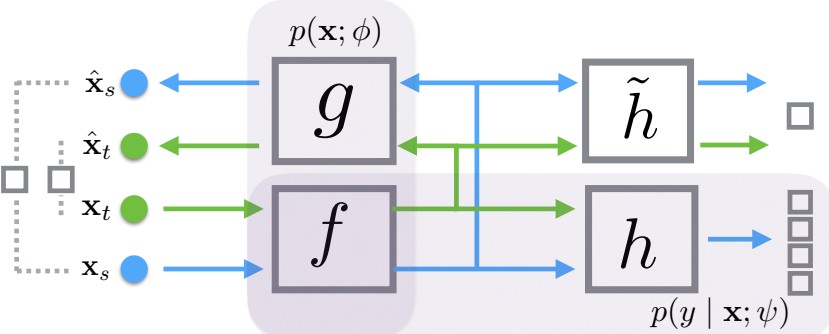

Figure 2: Model architecture of DAuto and its computation flow during learning phase. DAuto contains three components: the autoencoder ($f$ and $g$), the domain classifier ($\tilde{h}$) and the label predictor ($h$). The autoencoder instantiates the marginal distribution $p(\mathbf{x})$; the encoder $f$ and the label predictor instantiates the conditional probability $p(y \mid \mathbf{x})$. The domain classifier works as a regularizer. The model contains three different objectives: the domain classification loss, reconstruction loss as well as the label prediction loss. In the inference phase, only $f$ and $h$ will be used.

### 3.4 A CONCRETE MODEL (DAUTO)

We use neural networks as flexible function approximators for our desired transformations $f$ and $g$. Specifically, we use fully-connected neural networks to parametrize $f$ and $g$ and softmax function to parametrize $h$. If $y \in \mathbb{R}$, we can simply change the softmax function to be an affine function as the output. For the simplicity of discussion, assume we only use a one-layer fully connected network to represent $f$ and $g$: $f(\mathbf{x}) = \sigma(W_1\mathbf{x})$ and $g(\mathbf{z}) = \sigma(W_2\mathbf{z})$, where $W_1 \in \mathbb{R}^{D \times d}$, $W_2 \in \mathbb{R}^{d \times D}$ and $\sigma(\cdot)$

is an element-wise nonlinear activation function, e.g., the rectify linear unit. For ease of notation, we also omit the translation terms from above affine transformations. Let $h(\mathbf{z}) = \text{softmax}(W_h \mathbf{z})$ be the softmax layer to compute the conditional probability of class assignment.

Although our model has the capacity to learn the shared transformation $f$ under which unlabeled data from both domains have similar marginal distributions, the objective function discussed so far does not necessarily induce such a transformation. For the purpose of domain adaptation, it is necessary to add a regularizer that enforces this constraint. One popular and effective choice is the $\mathcal{A}$-distance introduced by (Kifer et al., 2004; Ben-David et al., 2007; 2010). It can be shown that the $\mathcal{A}$-distance can be approximated by the binary classification error of the domain classifier that discriminates instances from the source or the target domain (Ben-David et al., 2007). The intuition here is: given a fixed class of binary labeling functions, if there exists a function that is easy to tell instances in the source domain from those in the target domain, then the distance between these two domains is large. On the other hand, if the set of labeling functions are confused by such task, then we can think the distance between these two domains is small. Following DANN, here we take the same approach: let $\tilde{h} = \text{softmax}(W_d \mathbf{z})$ be the domain classifier where $\mathbf{z} = \sigma(W_1 \mathbf{x})$ is the shared representation constructed by encoder $f$. The regularizer takes the form as a convex surrogate loss for the binary 0-1 error. A common choice is the cross-entropy loss. Putting all together, the optimization problem of our joint model is given by:

$$\min_{W_1, W_2, W_h} \max_{W_d} \quad \sum_{i=1}^m \mathcal{L}_y(\mathbf{x}_i, y_i; W_1, W_h) + \lambda \sum_{j=1}^n \mathcal{L}_r(\mathbf{x}_j; W_1, W_2) - \mu \sum_{j=1}^n \mathcal{L}_d(\mathbf{x}_j; W_1, W_d) \quad (8)$$

where $\mathcal{L}_y(\cdot, \cdot)$ is the prediction loss, $\mathcal{L}_r(\cdot)$ is the reconstruction loss and $\mathcal{L}_d(\cdot)$ is the domain classification loss. We illustrate the model architecture in Fig. 2. As we show above, the cross-entropy loss for the learning task and the reconstruction loss are essentially the negative joint log-likelihoods of both labeled instances and unlabeled instances. The domain classification loss works as a regularizer to incorporate our prior knowledge that the encoder $f$ should form an invariant transformation. As a result, DAuto is designed to achieve the following three objectives simultaneously in a unified framework: 1). It learns representations that are informative for the main learning task in the source domain. 2). It learns robust representations under reconstruction loss. 3). It learns domain-invariant features that are indistinguishable between the source and the target domains.

## 4 EXPERIMENTS

We first evaluate DAuto on synthetic experiments with MNIST, and then compare it with state-of-the-art models, including mSDA, the Ladder network, DANN and ADDA. We report experimental results on the Amazon benchmark dataset, three digit datasets (MNIST, SVHN and USPS) and a large-scale time-series dataset for speech recognition.

### 4.1 DATASETS AND EXPERIMENTAL SETUP

**Synthetic experiment with MNIST**. The experiment contains 4 tasks: each task is a binary classification problem on judging whether the given image is a specific number or not. We choose 4 digits for these tasks: 3, 7, 8 and 9, because 9 and 7 are similar in many handwritten images while 3 and 9 (7 and 8) are quite different. Clearly domain adaptation is not always possible. We would like to verify that DAuto succeeds when two domains are close to each other, and also show a failure case when domains are sufficiently different. To do so, for each task on recognizing digit $i \in \{3, 7, 8, 9\}$, we sample 1000 images from the training set, of which 500 are digit $i$ and the others are digits not in $\{3, 7, 8, 9\}$; we sample $\sim$1,500 images from the original test set, of which $\sim$750 images are digit $i$ and the others are digits not in $\{3, 7, 8, 9\}$. There are 16 pairs of experiments altogether for each possible pair of $i, j \in \{3, 7, 8, 9\}$ as source and target domains. We design a well-controlled experiment to compare DAuto with a standard multilayer perceptron (MLP) and DANN: all algorithms share the same network structure. Also, we apply the same training procedure to all algorithms so that the difference in performance can only be explained by the additional domain regularizer as well as the reconstruction loss in DAuto.

**Multi-class Classification**. DAuto can be easily applied in multi-class classification as well. To see this, we test our model on MNIST, USPS and SVHN, all of which contain images of 10 digits.

MNIST contains 60,000/10,000 train/test instances; USPS contains 7,291/2,007 train/test instances, and SVHN contains 73,257/26,032 train/test instances. Before training, we preprocess all the instances into gray scale single-channel images of size 16x16, so that they can be used by the same network. Again, we perform a controlled experiment to ensure a fair comparison between the evaluated models. The network structures are exactly the same for all the approaches: 2 hidden layers with 1024, 512 units, following by a softmax output layer with 10 units. The domain discriminators in all the models are chosen to be logistic regression. During training, both batch size (400) and dropout rate (0.3) are also fixed to be same among all the experiments. Hence again, the difference in performance can only be explained by the different objective functions of different models.

**Sentiment Analysis**. The Amazon dataset consists of reviews of products on Amazon (Blitzer et al., 2007). The task is to predict the polarity of a text review, i.e., whether the review for a specific product is positive or negative. The dataset contains text reviews for the following four categories: books (B), DVDs (D), electronics (E), and kitchen appliances (K). Each of the product contains 2000 text reviews as training data, and 4465 (B), 3586 (D), 5681 (E), 5945 (K) reviews as test data. Each text review is described by a feature vector of 5000 dimensions, where each dimension correspond to a word in the dictionary. The dataset is a benchmark data that has been frequently used for the purpose of sentiment analysis (Blitzer et al., 2006; 2007; Chen et al., 2012; Ajakan et al., 2014; Ganin & Lempitsky, 2015; Long et al., 2016). For each source-target pair, we train the corresponding models completely on labeled source instances with access to unlabeled target instances. We use the classification accuracy in the target domain as our main metric.

**Speech recognition**. DAuto can be naturally extended to time-series modeling. In this experiment we apply DAuto to speech recognition, where a recurrent neural network is trained to ingest speech spectrograms and generate text transcripts. The model we use is a variant of DeepSpeech 2 (Amodei et al., 2015), which is composed of 1 convolution layer followed by 3 stacked bidirectional LSTM layers, and one fully connected layer before a softmax layer. At each time step, the input to the network is a log spectrogram feature, and the output is a character or blank. The network is trained end-to-end using the connectionist temporal classification (CTC) loss (Graves et al., 2006), which is the negative log-likelihood of training utterances. To extend DAuto in sequential models, besides the global CTC loss, we regularize it at each time step with both the reconstruction loss from the autoencoder as well as the adversarial loss from the domain classifier. We evaluate DAuto and compare it with other algorithms for an adaptation task across three different accented datasets, of which one is recorded from native English speakers, and the other two are recorded from speakers with Mandarin and Indian accents, respectively. Each dataset contains $\sim$33 hours of labeled audio data from $\sim$25,000 user utterances, and we randomly sample 80 percent of them as training set and the rest are used as test set.

We compare DAuto with the following methods: 1. **No-Adapt**. This is the baseline model which ignores the possible shifts between domains. In order to make the comparison as fair as possible, in all our experiments DAuto shares the same prediction model with No-Adapt. 2. **mSDA**. mSDA pretrains all the unlabeled instances from both the source and the target domains to build a feature map for the input space. The constructed representations from mSDA are used to train a SVM classifier as suggested in the original paper (Chen et al., 2012). In all the experiments, we set the corruption level to be 0.5 in training mSDA, and stack the same number of layers of autoencoders as in DAuto. 3. **Ladder Network (Ladder)**. The Ladder network (Rasmus et al., 2015) is a novel structure aiming for semi-supervised learning. It is a hierarchical denoising autoencoder where reconstruction errors between each pair of hidden layers are incorporated into the objective function. 4. **DANN** and 5. **ADDA**. Again, we use exactly the same inference structure for DANN and ADDA as in No-Adapt, Ladder, and DAuto. For all the experiments, we use early-stop to avoid overfitting. We implement all the models and ensure that all the preprocessing for data are the same for all the algorithms, so that the differences in experimental results can only be explained by the differences in models themselves. We defer detailed description about models used in each experiment to the supplementary material.

## 4.2 Results and Analysis

**MNIST**. We list the classification accuracies on 16 pairs of tasks using No-Adapt, DANN and DAuto in Table 1. Besides the 12 pairs of tasks under the domain adaptation setting, we also show 4 additional tasks where both the training and test sets are from the same domain. The scores from these 4 tasks can be used as empirical upper bounds to compare with the performance of domain

Table 1: Classification accuracies on 16 tasks between No-Adapt, DANN and DAuto. $S \to T$ means that $S$ is the source domain and $T$ is the target domain. Improvements over baselines are highlighted.

| Task | No-Adapt | DANN | DAuto | Task | No-Adapt | DANN | DAuto |
|------|----------|------|-------|------|----------|------|-------|
| $9 \to 9$ | 0.967 | 0.967 | 0.965 | $7 \to 9$ | 0.777 | 0.807 | **0.837** |
| $9 \to 7$ | 0.883 | 0.911 | **0.917** | $7 \to 7$ | 0.977 | 0.979 | 0.979 |
| $9 \to 8$ | 0.595 | 0.599 | **0.655** | $7 \to 8$ | 0.519 | 0.524 | 0.540 |
| $9 \to 3$ | 0.532 | 0.539 | **0.686** | $7 \to 3$ | 0.523 | 0.514 | 0.553 |
| $8 \to 9$ | 0.535 | 0.547 | **0.640** | $3 \to 9$ | 0.535 | 0.643 | **0.734** |
| $8 \to 7$ | 0.591 | 0.650 | **0.705** | $3 \to 7$ | 0.738 | 0.795 | **0.806** |
| $8 \to 8$ | 0.965 | 0.963 | 0.957 | $3 \to 8$ | 0.612 | 0.671 | **0.737** |
| $8 \to 3$ | 0.697 | 0.656 | 0.698 | $3 \to 3$ | 0.973 | 0.976 | 0.973 |

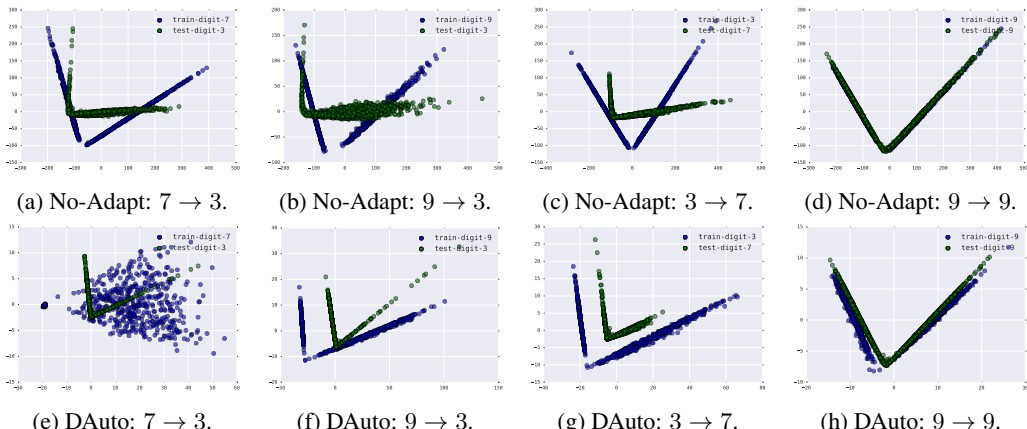

(a) No-Adapt: $7 \to 3$.  (b) No-Adapt: $9 \to 3$.  (c) No-Adapt: $3 \to 7$.  (d) No-Adapt: $9 \to 9$.

(e) DAuto: $7 \to 3$.  (f) DAuto: $9 \to 3$.  (g) DAuto: $3 \to 7$.  (h) DAuto: $9 \to 9$.

Figure 3: 2 dimensional PCA embeddings of learned representations with and without DAuto adaptation. All the representations are extracted from the last hidden layer of the networks. Blue and green dots represent instances from the labeled training set and unlabeled target set, respectively.

adaptation algorithms. DAuto significantly improves over the No-Adapt baseline in 10 out of the total 12 possible pairs, showing that it indeeds has the desired capability for domain adaptation.

On the other hand, we would also like to highlight that *not* all domain adaptation tasks are successful: the prediction accuracies of all the three algorithms on task $7 \to 8$ and $7 \to 3$ are only marginally above random guesses (0.5). When the digits are similar, we observe very successful domain transfer using DAuto even the algorithm does not see any instances from the target category during training. To qualitatively study both the successful and failure cases, we project both representations with and without DAuto adaptation onto 2 dimensional space using PCA, shown in Fig. 3. Several interesting observations can be made from Fig. 3: when domain adaptation is successful (Fig. 3g, 3f), the principal directions of learned representations from both domains are well-aligned with each other; on the other hand, when adaptation fails (Fig. 3e), representations do not share the same principal directions. As a special case, when the source and target domains share the same distribution (Fig. 3h), DAuto still works and will not degrade.

**Multi-class Classification**. The results on multi-class classification of digits are shown in Table 2, where we highlight the successful domain adaptations using green colors and failure cases using red colors. The datasets in row correspond to the source domains and those in columns correspond to the target domains. DANN, ADDA and DAuto all contain one failure case (USPS→SVHN, USPS→SVHN and MNIST→USPS), which may be explained by the intrinsic difference between the SVHN datasets and the other two. However, on those three datasets, whenever both DANN and DAuto succeed in domain adaptation, DAuto usually outperforms DANN by around 2 percent accuracy. On the other hand, ADDA achieves far better result on MNIST→USPS than both DANN and DAuto, while it also performs worse than DAuto on SVHN→MNIST and USPS→MNIST. Note that in this experiment all the methods share exactly the same experimental protocol, hence the

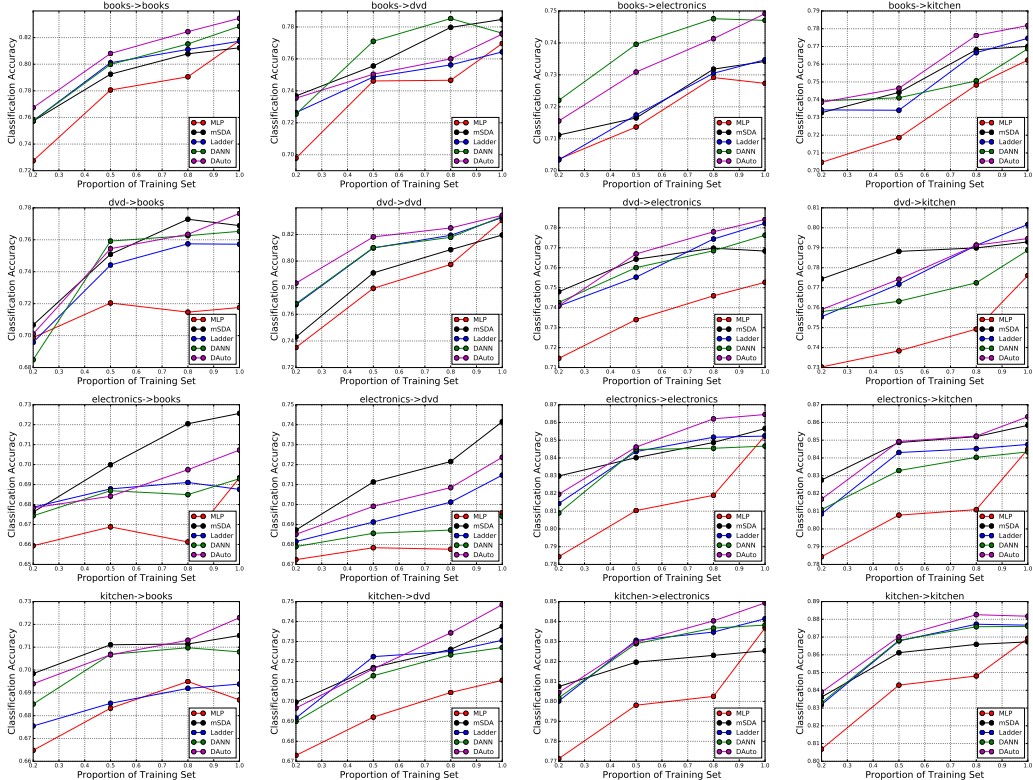

Figure 4: Test set performances of MLP, Ladder, mSDA, DANN and DAuto with increasing training set sizes: from 0.2 to 1.0. DAuto achieves the best accuracy on 12 out of 16 tasks.

Table 2: Classification accuracy on the digits experiment from No-Adapt, DANN, ADDA and DAuto. Improvements over baseline method are highlighted in green, and decreases in performance are shown in red. Table best viewed in color.

| | No Adapt | | | DANN | | |
|---|---|---|---|---|---|---|
| | **SVHN** | **MNIST** | **USPS** | **SVHN** | **MNIST** | **USPS** |
| **SVHN** | 0.8553 | 0.5459 | 0.5277 | 0.8596 | 0.5690 | 0.5426 |
| **MNIST** | 0.2054 | 0.9883 | 0.6442 | 0.2241 | 0.9880 | 0.6500 |
| **USPS** | 0.1628 | 0.3396 | 0.9507 | 0.1585 | 0.3562 | 0.9517 |
| | **ADDA** | | | **DAuto** | | |
| | **SVHN** | **MNIST** | **USPS** | **SVHN** | **MNIST** | **USPS** |
| **SVHN** | 0.8707 | 0.5542 | 0.5561 | 0.8626 | 0.5864 | 0.5655 |
| **MNIST** | 0.2091 | 0.9894 | 0.6856 | 0.2086 | 0.9869 | 0.6428 |
| **USPS** | 0.1602 | 0.3570 | 0.9512 | 0.1717 | 0.3762 | 0.9537 |

difference can only be explained by their different objective functions and model designs. In other words, the adaptation can benefit from the reconstruction error from autoencoders, which works as an unsupervised regularizer.

**Amazon**. To show the effectiveness of different domain adaptation algorithms when labeled instances are scarce, we evaluate the five algorithms on the 16 tasks by gradually increasing the size of the labeled training instances, but still use the whole test dataset to measure the performance. More specifically, we use 0.2, 0.5, 0.8 and 1.0 fraction of the available labeled instances from source domain during training. A successful domain adaptation algorithm should be able to take advantage of the unlabeled instances from the target domain to help generalization even when the amount of labeled instances available is small. We plot the results in Fig. 4: all the domain adaptation algorithms

Table 3: CTC loss on 9 tasks from LSTM (No-Adapt), DANN and DAuto. A lower loss indicates a better speech model. Improvements over baseline method are highlighted in green, and decreases in performance are shown in red. Table best viewed in color.

|  | LSTM (No Adapt) | | | DANN | | | DAuto | | |
|---|---|---|---|---|---|---|---|---|---|
|  | US | CN | IN | US | CN | IN | US | CN | IN |
| US | 263.7 | 160.4 | 408.9 | 189.3 | 112.4 | 428.9 | **185.9** | **97.9** | 486.1 |
| CN | 226.6 | 110.9 | 375.4 | 186.8 | 66.4 | 453.0 | **160.7** | **45.7** | 494.8 |
| IN | 389.7 | 245.5 | 376.5 | 498.4 | 429.1 | 244.7 | 493.0 | 470.3 | **241.3** |

are able to use the unlabeled instances from the target domain to help generalization. However, the differences between the baseline MLP and DAuto are getting smaller with the increase of the size of training data. This phenomenon supports our probabilistic framework showing that DAuto can effectively use the unlabeled data. We report the detailed numeric scores achieved by each algorithm and their statistical significance test in the supplementary material. In summary, DAuto outperforms all the other competitors on 12 out of the 16 tasks, which is consistently better than its competitors.

**Speech Recognition**. To avoid possible external error during the decoding process, which is usually not stable on noisy data, we directly report the CTC loss obtained from different algorithms. We have $3 \times 3$ pairs of domain adaptation tasks, where each source/target domain ranges from speech with {IN, CN, US} accents. We show the results in Table 3. Again, we observe decreases in performance when applying domain adaptation algorithms: the transfer between speech with Indian accents and the other two fails. However, when two domains are similar (US vs CN) where adaptation is possible, DAuto consistently outperforms DANN. We want to bring readers' attention that both DANN and DAuto improve over baseline method when training with unlabeled instances from the same domain (diagonal of the table). This means both DANN and DAuto have the effect of semi-supervised learning algorithms: when the source and the target domains are indeed same, both DANN and DAuto benefit from using unlabeled instances. From the generative interpretation of our probabilistic framework, DAuto achieves this goal by maximizing the marginal probability of unlabeled instances, and due to the shared component, this further helps training the discriminative model.

## 5 CONCLUSION

We propose a probabilistic framework that incorporates both generative and discriminative modeling in a principled way, which also helps us to interpolate between generative and discriminative extremes through a specific choice of prior distribution where a subset of model parameters is shared. The instantiated model, DAuto, allows us to use unlabeled instances from both domains in a principled way. This instantiation also shows that the empirical success of autoencoders in semi-supervised learning and domain adaptation can be explained as maximizing the marginal log-likelihoods of unlabeled data, where kernel density estimators are used to model the marginal distributions. This provides the first probabilistic justification for joint training with autoencoders in practice. Experimentally we show that DAuto can be successfully applied to domain adaptation problems, and has a natural extension to time series as well.

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

# A  MORE DETAILS ABOUT EXPERIMENTS

## A.1  MNIST SYNTHETIC EXPERIMENTS

The networks we used in this experiment contains 3 hidden layers, each contain 500, 200 and 100 hidden units, respectively. The input layer contains $28 \times 28 = 784$ units, and the output layer is a single unit parametrized by a logistic regression model. Except the output layer, all the hidden layers use ReLU as the nonlinear activation function. The same network structure is used for both DANN and DAuto. During training, we use AdaDelta as the optimization algorithm for all the models. For DAuto, both hyperparameters $\lambda$ and $\mu$ are chosen from the following range: $\lambda, \mu \in \{10^{-8}, 10^{-7}, \ldots, 10^2\}$, where we further partition the target domain set as development set and the held-out test set, combined with early stopping for model selection. For all the experiments, we fix the learning rate to be 1.0.

## A.2  AMAZON BENCHMARK DATASET

MLP, Ladder, DANN and DAuto all share the same network structure: the input layer contains 5,000 units, followed by a fully-connected hidden layer with 500 units and the output layer, which is a logistic regression model. Correspondingly, mSDA pretrains a two-layer stacked auto-encoder, hence the feature representation pretrained by mSDA in this experiment has 10,000 dimensions. Ladder and DAuto also use unlabeled instances to pretrain model parameters in an purely unsupervised way. Again, for all the neural network based models, we use ReLU as the nonlinear activation function and use AdaDelta as the optimization algorithm during training. For DAuto, both hyperparameters $\lambda, \mu \in \{10^{-4}, \ldots, 10^2\}$. Again, we fix the learning rate to be 1.0.

We report the classification accuracies on the test data of the 16 pairs of tasks to have a thorough comparisons among the 5 models in Table 4. DAuto outperforms all the other competitors on 12 out of 16 tasks, whereas DANN achieves the best test set accuracy on $D \rightarrow D$; Ladder scores highest on $D \rightarrow E$ and $D \rightarrow K$; mSDA performs the best on $E \rightarrow B$ and $E \rightarrow D$. We emphasize that DAuto performs consistently better than all the other competitors. Note that the only difference between DANN and DAuto is the autoencoder regularizer that forces the feature learning part to learn robust features, we conclude that DAuto successfully helps to build robust representations.

Table 4: Sentiment classification (positive/negative) accuracies on 16 tasks of 5 models: MLP, Ladder, mSDA, DANN and DAuto. $S \rightarrow T$ means that $S$ is the source domain and $T$ is the target domain.

| Task | MLP | mSDA | Ladder | DANN | DAuto |
|------|-----|------|--------|------|-------|
| B→B | 0.823 | 0.812 | 0.817 | 0.828 | **0.834** |
| B→D | 0.770 | **0.785** | 0.764 | 0.773 | 0.776 |
| B→E | 0.728 | 0.734 | 0.735 | 0.734 | **0.749** |
| B→K | 0.762 | 0.770 | 0.775 | 0.769 | **0.782** |
| D→B | 0.768 | 0.769 | 0.757 | 0.765 | **0.776** |
| D→D | 0.830 | 0.820 | 0.832 | **0.834** | **0.834** |
| D→E | 0.753 | 0.768 | **0.784** | 0.776 | **0.784** |
| D→K | 0.776 | 0.793 | **0.802** | 0.789 | 0.795 |
| E→B | 0.693 | **0.726** | 0.683 | 0.693 | 0.707 |
| E→D | 0.696 | **0.741** | 0.715 | 0.694 | 0.724 |
| E→E | 0.850 | 0.857 | 0.854 | 0.847 | **0.864** |
| E→K | 0.845 | 0.858 | 0.848 | 0.843 | **0.863** |
| K→B | 0.687 | 0.715 | 0.694 | 0.710 | **0.723** |
| K→D | 0.711 | 0.738 | 0.731 | 0.727 | **0.748** |
| K→E | 0.838 | 0.825 | 0.843 | 0.838 | **0.849** |
| K→K | 0.869 | 0.867 | 0.874 | 0.873 | **0.882** |

To check whether the accuracy differences between those five methods are significant or not, we perform a paired $t$-test and report the two-sided $p$-value under the null hypothesis that two related

paired samples have identical means. We show the $p$-value matrix in Table 5 where for each pair of methods, we report the mean $p$-value among the 16 tasks.

Table 5: $p$-value matrix between different domain adaptation algorithms. Each entry in the matrix corresponds to the mean $p$-value under a paired $t$-test on 16 domain adaptation tasks.

|         | MLP   | mSDA  | Ladder | DANN  | DAuto |
|---------|-------|-------|--------|-------|-------|
| **MLP**    | -     | 0.147 | 0.181  | 0.460 | 0.025 |
| **mSDA**   | 0.147 | -     | 0.136  | 0.154 | 0.175 |
| **Ladder** | 0.181 | 0.136 | -      | 0.273 | 0.225 |
| **DANN**   | 0.460 | 0.154 | 0.273  | -     | 0.072 |
| **DAuto**  | 0.025 | 0.175 | 0.225  | 0.072 | -     |

From Fig. 4, we can also observe that given a small fraction of data, there are large gaps between DAuto and the MLP baseline on most of the tasks. Interestingly, these gaps become smaller on certain tasks, e.g., kitchen->electronics, books->kitchen. Besides, we observe a gap consistently as the fraction of training data increases. This convinces our conjecture that DAuto is more effective than the MLP baseline in learning informative representations from low-resource data, and has the semi-supervised learning effect due to its generative nature.

### A.3 SPEECH RECOGNITION WITH ACCENTS

The model structure of the recurrent network we used for this experiment is as follows: at each time step, the input feature is a log spectrogram feature of 161 dimensions. This is followed by a 1D convolution layer with 1024 kernels, and 3 stacked bidirectional LSTM layers, each of which contains 1024 hidden units. The output of the last bidirectional LSTM layer is connected to a fully-connected softmax layer with 29 outputs, which represents 26 characters and three special characters: space, apostrophe and blank. We use the publicly available CTC loss implementation `https://github.com/baidu-research/warp-ctc` and our experiments are performed under the public codebase: `https://github.com/baidu-research/ba-dls-deepspeech`.

