# OpenReview forum: "Principled Hybrids of Generative and Discriminative Domain Adaptation"
_ICLR.cc/2018/Conference — Reject_

### Official Review · AnonReviewer3 · 2017-11-27
**Different angle to explain the DA methods, but the novelty is insufficient**

**Rating:** 5
**Confidence:** 4

**Review:**

This paper proposed a probabilistic framework for domain adaptation that properly explains why maximizing both the marginal and the conditional log-likelihoods can achieve desirable performances.
However, I have the following concerns on novelty.

1. Although the paper gives some justiification why auto-encoder can work for domain adaptation from perspective of probalistics model, it does not give new formulation or algorithm to handle domain adaptation.  At this point, the novelty is weaken.
2. In the introduction, the authors mentioned “limitations of mSDA is that it needs to explicitly form the covariance matrix of input features and then solves a linear system, which can be computationally expensive in high dimensional settings.” However, mSDA cannot handle high dimension setting by performing the  reconstruction with a number of  random non-overlapping sub-sets of input features. It is not clear why mSDA cannot handle time-series data but DAuto can.  DAuto does not consider the sequence/ordering of data either.
3. If my understanding is not wrong, the proposed DAuto is just a simple combination of three losses (i.e. prediction loss, reconstruction loss, domain difference loss). As far as I know, this kind of loss is commonly used in most existing methods.

---

> ### Author Response · Authors · 2017-12-14
> **Response to Reviewer 3**
>
> We'd like to thank the reviewer for the thoughtful questions, and we want to address them here.
>
> Q: Although the paper gives some justiification why auto-encoder can work for domain adaptation from perspective of probabilistic model, it does not give new formulation or algorithm to handle domain adaptation.  At this point, the novelty is weaken.
>
> Besides providing a probabilistic justification for autoencoders, we also propose DAuto as a new model to handle domain adaptation. At the end DAuto boils down to DANN+autoencoders, and to the best of our knowledge we didn't see any existing work using this combination of structures for domain adaptation, which we treat as a novel contribution.
>
>
> Q: However, mSDA cannot handle high dimension setting by performing the reconstruction with a number of random non-overlapping sub-sets of input features. It is not clear why mSDA cannot handle time-series data but DAuto can. DAuto does not consider the sequence/ordering of data either.
>
> Yes, we agree with the reviewer that performing the reconstruction by random non-overlapping subsets of features could serve as an approximation to the solution of the original linear system that mSDA attempts to solve. Perhaps we could phrase it as "One of the limitations of mSDA is that it needs to explicitly form the covariance matrix of input features and then solves a linear system, which can be computationally expensive to solve exactly in high dimensional settings, but approximate scheme exists."
>
> What we would like to emphasize is the fact that mSDA is more like a two-stage algorithm, where feature learning and discriminative training are separated. On the other hand, DAuto is more end-to-end, which combines feature learning, discriminative training and domain adaptation in a unified model. In time-series modeling the latter might be more favorable because it has the power to adapt to the change in distribution along time. We've removed the sentence "On the other hand, it is not clear how to extend mSDA so that it can also be applied for time-series modeling." from related work.

---

### Official Review · AnonReviewer2 · 2017-11-27
**An enjoyable paper, with some room for improvements**

**Rating:** 6
**Confidence:** 3

**Review:**

This is a very well-written paper that shows how to successfully use (generative) autoencoders together with the (discriminative) domain adversarial neural network (DANN) of Ganin et al.
The construction is simple but nicely backed by a probabilistic analysis of the domain adaptation problem.

The only criticism that I have towards this analysis is that the concept of shared parameter between the discriminative and predictive model (denoted by zeta in the paper) disappear when it comes to designing the learning model.

The authors perform numerous empirical experiments on several types of problems. They successfully show that using autoencoder can help to learn a good representation for discriminative domain adaptation tasks. On the downside, all these experiments concern predictive (discriminative) problems. Given the paper title, I would have expected some experiments in a generative context. Also, a comparison with the Generative Adversarial Networks of Goodfellow et al. (2014) would be a plus.
I would also like to see the results obtained using DANN stacked on mSDA representations, as it is done in Ganin et al. (2016).

Minor comments:
- Paragraph below Equation 6:  The meaning of $\phi(\psi)$ is unclear
- Equation (7): phi and psi seems inverted
- Section 4: The acronym MLP is used but never defined.

=== update ===
I lowered my score and confidence, see my new post below.

---

> ### Author Response · Authors · 2017-12-14
> **Response to Reviewer 2**
>
> Thanks for the accurate summarization and all the comments! We attempt to clarify several points below.
>
> Q: The only criticism that I have towards this analysis is that the concept of shared parameter between the discriminative and predictive model (denoted by zeta in the paper) disappear when it comes to designing the learning model.
>
> zeta does not disappear in DAuto, and in fact DAuto is precisely designed to follow the probabilistic principle discussed in Section 3.1 to 3.3. In Fig. 2, the zeta corresponds to the shared component f: f and g together instantiate p(x), while f and h together instantiate p(y|x). In other words, phi = parameters in f union parameters in g, and psi = parameters in f union parameters in h. So zeta = phi intersects psi = parameters in f.
>
>
> Q: Also, a comparison with the Generative Adversarial Networks of Goodfellow et al. (2014) would be a plus.
>
> This is indeed a great question, and we'd like to clarify here. Although DAuto is designed to consider the marginal distribution over x, and we use kernel density estimation to construct this density explicitly, DAuto does not have efficient sampling schemes to actually generate samples from the distribution. On the other hand, from GAN we cannot compute the density of a given instance explicitly, but it allows us to draw samples from it in a straightforward way.
>
>
> Q: I would also like to see the results obtained using DANN stacked on mSDA representations, as it is done in Ganin et al. (2016).
>
> Stacking with mSDA representations can indeed boost the performances of all the models, including DANN, Ladder and DAuto uniformly on the Amazon dataset. On average we see around 2 percent improvements in classification accuracy on Amazon.
>
>
> Thanks again for all the detailed comments as well! We've updated them in the revised version.

---

> > ### Comment · AnonReviewer2 · 2018-01-12
> > **On second thought paper, the paper might not be novel enough.**
> >
> > The other reviews made me realize that the contribution can be improved. Consequently, I lowered my score from 7 to 6.
> >
> > I honestly don't remember what I was thinking when I wrote my comment about the zeta term. The authors made it clear in their rebuttal that there is a direct correspondence between this and the shared parameters of the network. However, there is clearly room to analyze more the influence of the proportion of shared parameters in the model. Currently, the probabilistic framework says that we need those to some extent, and the experiments use a fix model without much explanation.

---

### Official Review · AnonReviewer1 · 2017-11-29
**A probabilistic framework for domain adaptation, some more recent baselines missing**

**Rating:** 5
**Confidence:** 4

**Review:**

The authors propose a probabilistic framework for semi-supervised learning and domain adaptation. By varying the prior distribution, the framework can incorporate both generative and discriminative modeling.  The authors emphasize on one particular form of constraint on the prior distribution, that is weight (parameter) sharing, and come up with a concrete model named Dauto for domain adaptation. A domain confusion loss is added to learn domain-invariant feature representations. The authors compared Dauto with several baseline methods on several datasets and showed improvement.

The paper is well-organized and easy to follow. The probabilistic framework itself is quite straight-forward. The paper will be more interesting if the authors are able to extend the discussion on different forms of prior instead of the simple parameter sharing scheme.

The proposed DAuto is essentially DANN+autoencoder.  The minimax loss employed in DANN and DAuto is known to be prone to degenerated gradient for the generator. It would be interesting to see if the additional auto-encoder part help address the issue.

The experiments miss some of the more recent baseline in domain adaptation, such as Adversarial Discriminative Domain Adaptation (Tzeng, Eric, et al. 2017).

It could be more meaningful to organize the pairs in table by target domain instead of source, for example, grouping 9->9, 8->9, 7->9 and 3->9 in the same block. DAuto does seem to offer more boost in domain pairs that are less similar.

---

> ### Author Response · Authors · 2017-12-14
> **Response to Reviewer 1**
>
> Thank you for providing thoughtful comments and suggestions. We attempt to answer the questions below.
>
> Q:  The paper will be more interesting if the authors are able to extend the discussion on different forms of prior instead of the simple parameter sharing scheme.
>
> As we show in section 3.1, one necessary condition under the probabilistic framework is that phi and psi cannot be independent, otherwise the maximization of marginal distribution over x would be independent of the discriminative task, hence phi and psi need to be correlated. While there are many ways that phi and psi could be correlated, one sufficient and frequently used assumption in practice is that they share some common parameters, under which we show that this corresponds to minimizing reconstruction loss using autoencoders. Other possible choices include phi being a function of psi, etc., but such choice has not been frequently used in practice, so we focus on the parameter sharing scheme in this work.
>
>
> Q:  The minimax loss employed in DANN and DAuto is known to be prone to degenerated gradient for the generator. It would be interesting to see if the additional auto-encoder part help address the issue.
>
> Yes, that's right. As we observed empirically in our experiments the reconstruction loss from autoencoders do indeed tend to stablize the joint optimization, but theoretically we don't know how to formally prove this yet.
>
>
> Q:  The experiments miss some of the more recent baseline in domain adaptation, such as Adversarial Discriminative Domain Adaptation (Tzeng, Eric, et al. 2017).
>
> Thanks for pointing out this work! We've incorporated a comparison to ADDA as well, please check the revised submission. As a high-level summary, given that all the methods use exactly the same network structures and training protocols, ADDA does perform very well on several adaptation tasks, while DAuto excels on other tasks. This shows that using asymmetric encoders and using reconstruction loss as regularization can both contribute to adaptation, perhaps in an orthogonal way.

---

### Author Response · Authors · 2017-12-14
**Thanks for all the comments and suggestions**

We thank all the reviewers for the time devoted to provide thoughtful comments and suggestions. We attempt to answer the questions separately below.

---

### Decision · Program_Chairs · 2018-01-29
**ICLR 2018 Conference Acceptance Decision**

**Decision:**

Reject

**Comment:**

Pros
-- Nice way to formulate domain adaptation in a Bayesian framework that explains why autoencoder and domain difference losses are useful.

Cons
-- Model closely follows the framework, but the overall strategy is similar to previous models (but with improved rationale).
-- Experimental section can be improved. It would interesting to explore and develop the relationship between the proposed technique and Tzeng et al.

Given the aforementioned cons, the AC is recommending that the paper be rejected.